# Higher Academic Stress Was Associated with Increased Risk of Overweight and Obesity among College Students in China

**DOI:** 10.3390/ijerph17155559

**Published:** 2020-07-31

**Authors:** Yonghua Chen, Xi Liu, Ni Yan, Wanru Jia, Yahui Fan, Hong Yan, Lu Ma, Le Ma

**Affiliations:** 1School of Public Health, Xi’an Jiaotong University Health Science Center, Xi’an 710061, China; chyh_2006@mail.xjtu.edu.cn (Y.C.); lx2142008010@stu.xjtu.edu.cn (X.L.); n19940522@stu.xjtu.edu.cn (N.Y.); jiawanru123@stu.xjtu.edu.cn (W.J.); huikai18@stu.xjtu.edu.cn (Y.F.); yanhonge@mail.xjtu.edu.cn (H.Y.); 2Research Centre on College Students Ideological Education and Practice, Xi’an Jiaotong University, Xi’an 710061, China; 3Global Health Institute, School of Public Health, Xi’an Jiaotong University Health Science Center, Xi’an 710061, China

**Keywords:** overweight, obesity, college students, academic stress, negative learning events

## Abstract

This study examined associations between academic stress and overweight and obesity, and moderation effects of gender, grade, and types of college on such associations. Data on academic stress, negative learning events, weight, and height were self-reported by 27,343 college students in China in 2018. About 23% and 91% of students perceived high academic stress and suffered from at least one negative learning event during the past six months, respectively, especially for females, undergraduates, and students major in humanities and social science subject groups. Perceived academic stress was associated with increased risk of overweight and obesity among all students (OR = 1.05, 95%CI: 1.00–1.10), male (OR = 1.09, 95%CI: 1.03–1.15), undergraduate (OR = 1.06, 95%CI: 1.00–1.11), and students from subordinate universities (OR = 1.13, 95%CI: 1.01–1.26). Negative learning events were associated with increased risk of overweight and obesity among all students (OR = 1.05, 95%CI: 1.01–1.09), undergraduates (OR = 1.05, 95%CI: 1.01–1.09), and students from local universities (OR = 1.07, 95%CI: 1.00–1.14). Interventions are needed to reduce the high academic stress of college students, considering the modifying effects of gender, grade, and college type. Such interventions may further contribute to the prevention of overweight and obesity among college students.

## 1. Introduction

Youth overweight and obesity has become a serious public health concern, and the prevalence has been increasing at an alarming rate especially in developing countries [1]. Among Chinese college students, 22.7% of males and 8.4% of females aged 19–22 years were overweight or obese [2]. Excess weight in college students could increase the risk of developing physical (e.g., type 2 diabetes, hypertension) [3] and psychological health problems (e.g., depression and weight stigma) [4].

High levels of psychological stress have numerous deleterious effects on academic (e.g., impeding learning abilities), physical, and mental health outcomes among university students [5,6]. Psychological stress has been found to be a risk factor of overweight and obesity through multiple interacting biological and behavioral pathways [7]. A meta-analysis of 14 longitudinal studies showed that stresses including general life stress (caregiver stress, major life events) and job strain were positively associated with risk of obesity, albeit with a modest effect size [8]. Furthermore, previous studies suggested that the associations varied by socio-demographic factors. For example, higher perceived life stress was associated with overweight or obesity only among male students [9]. Previous study has indicated that leaving home to attend a post-secondary school, itself, is an important reason for stress, and compared with graduates, the transition mainly affected the freshmen and caused the series of adverse outcomes [10]. Furthermore, for the students in different majors, especially medical, a large number of studies have shown their high level of stress and the effect on both physical health and mental health [11,12].

As one of the most common chronic stressors for college students, prior research has indicated that academic stress may impair self-control and deteriorate health behaviors such as changing the dietary pattern [13], in turn, increasing the risk of overweight and obesity. For example, a laboratory study reported that college students took many more calories, carbohydrates, and sugars when academically stressed [14]. However, no study has focused on the effects of academic stress and negative learning events on overweight and obesity among college students. More and more college students were reported to be plagued with crippling bouts of academic stress, which refers to stress that occurs in the educational field, such as the worrying of the difficult tasks and being assessed by various tests [15], including perceived academic stress and the experience of negative learning events. In China, students’ academic excellence has been a social criterion and even become the only standard for Chinese parents to judge their children [16]. Rapid social changes and the implementation of the one-child policy have contributed to elevate academic and job competition and directly increase college student’s academic pressure [17]. It was indicated that the Chinese college students were confronting the highest level of academic stress compared with their counterparts in Japan and Korea, and tended to report more school-related negative events than those in United States [18,19]. However, no study has investigated the academic stress or negative learning events in Chinese college students until now.

With a large-scale sample of Chinese college students, this study aimed to investigate the associations between academic stress (perceived academic stress and negative learning events) and overweight and obesity. Furthermore, we also examined the potential modifying effects of gender, grade, and college type on such associations. We hypothesized that higher perceived academic stress and negative learning events experiences were associated with increased overweight and obesity among college students in China. Findings would provide insights to design interventions targeting academic stress to fight the obesity epidemic among college students.

## 2. Materials and Methods

### 2.1. Study Design and Participants

A total of 30,000 students participated in this cross-sectional study in 2018, of which 28,000 were undergraduates and 2000 were graduates. Stratified multilevel cluster random sampling was used to recruit participants at 30 colleges in Shan’xi Province, stratified by college types (subordinates, local, private, and vocational universities). Specifically, 800 undergraduates (200 for each grade from 1 to 4) and 200 graduates were invited in 4 subordinate and 6 local universities, respectively. In the other 6 local colleges and 4 private universities, 1000 undergraduates were recruited, where 300 students were randomly selected from grade 1 and grade 2, while 200 students were selected from grade 3 and grade 4, respectively. The number of the students in the range from grade 1 to 3 who participated in the project were 400, 300, and 300 in vocational universities. All students in the sampled grades were invited to complete the self-administered paper questionnaire in classroom settings in the absence of teachers being supported by a research assistant. It took about 20 min to complete the questionnaire. Trained and experienced research assistants briefed the participants about the study and questions in the questionnaire. Socio-demographic information, perceived academic stress, negative learning events, height, and body weight were used for data analyses. A sample of 27,343 participants (43.5% were boys) with complete key variables data were included in the final data analyses (missing rate = 8.86%).

Consent was sought from the school principals before the survey. It was announced that return of the completed questionnaire implied informed consent by students. No incentive was provided to the participants. The study was approved by the ethical committee of Xi’an Jiao Tong University, bearing registration number 2017-788 on 11 November 2017.

### 2.2. Variables and Measurements

#### 2.2.1. Outcomes

Overweight and obesity: BMI was calculated based on self-reported body weight (kg) divided by self-reported height (m^2^) squared. Overweight and obesity were defined based on the standard recommended by the Working Group for Obesity in China for Chinese adults (Normal weight: 18.5 kg/m^2^ < BMI ≤ 23.9 kg/m^2^, overweight: 24.0 kg/m^2^ ≤ BMI ≤ 27.9 kg/m^2^, Obese: BMI ≥ 28.0 kg/m^2^) [20].

#### 2.2.2. Exposure Variables

The exposure variables were perceived academic stress and negative learning events. Perceived academic stress was measured by one self-reported item “How much academic stress do you feel in the study?” using a 5-point Likert scale where 1 = “No”, 2 = “relatively low”, 3 = “average/general”, 4 = “relatively high” and 5 = “extremely heavy”, with a higher score indicating more perceived academic stress. The response categories were categorized into three categories in the descriptive analyses: low stress including “no = 1” or “relatively light = 2”, medium stress including “average/general = 3”, and high stress including “relatively high = 4” and “extremely heavy = 5”.

Negative learning events were measured by asking the participants whether the four negative learning events happened in the past six months (Response: yes or no). The four negative learning events included: “Failure in the exam or unsatisfactory performance”, “Heavy study load”, “family financial difficulties”, and “Pressure for further study”. Students who answered that these events happened in the past 6 months were asked to rate the degree to which these events affected them using a 5-point Likert scale from 1 = “never” to 5 = “extremely heavy”. The average scores of four events were calculated to indicate the negative impacts of the negative learning events, with higher score indicating more negative impacts.

#### 2.2.3. Covariates

Covariates included gender; major (Science and technology, Medical science, Agronomy, Humanities and social science, Economics and management, and Sports and art); college type (Subordinate universities, Local universities, Private universities, and Vocational universities); grades (undergraduate and graduate); academic attainment (below average, average, and above average); and the characteristics of their parents: paternal and maternal education level (≤junior middle school, senior middle school/vocational schools, and ≥college), and household income (<40,000/year, 50,000–100,000/year, and >100,000/year).

### 2.3. Statistical Analysis

Descriptive statistics were calculated. A Chi-square test (for categorical variables) and *t*-tests (for continuous variables) were conducted to test for the differences of perceived academic stress and the impacts of negative learning events across gender, grade, major, and college types.

Linear/logistic mixed-effects models were used to examine the associations of academic stress and negative learning events with overweight and obesity after adjusting for covariates using enter procedure. Effect sizes were presented as a beta coefficient with a standard error or odds ratio and a 95% confidence interval (95% CI). Stratified analyses were conducted by gender, grade, and college types to examine the potential modifying effects. Mixed-effects modeling was used to account for clustering effects. Stata 14 (StataCorp, College Station, TX, USA) was used in data analysis Statistical significance was set at *p* < 0.05.

## 3. Results

### 3.1. Characteristics of Participants

The basic characteristics of the 27,343 included college students are shown in Table 1 (*n* = 11,888 boys (43.4%) and *n* = 15,295 girls). About 13% of them were overweight or obese. Boys were more likely to be overweight and obese than girls (boys vs. girls: 20.8% vs. 7.3%). Most of them (93.4%) were undergraduate students. Most of the students majored in science and technology, and boys were more likely to study science and technology, while girls were more likely to study humanities and social science. About 90% of the students reported their academic attainment as average or above average. The majority of students’ fathers (83.6%) and mothers (86.5%) only attained junior middle school or lower education, and had a household income <40,000 per year (80.8%).

### 3.2. Characteristics of Perceived Academic Stress and Negative Learning Events among College Students

The prevalence of perceived high academic stress was 22.9%. Girls (23.3%) or students who study humanities and social science (26.5%) were more likely to report high perceived academic stress compared with boys (22.5%) and other major students (proportions ranged from 20.2% to 23.6%), respectively (Table 2).

Regarding experiencing negative learning events, about 91% of the college students experienced negative learning events in the past 6 months. Females (91.9%), undergraduate students (91.1%), and students studying humanities and social science (92.1%) were more likely to experience negative learning events compared with males (89.4%), graduate students (87.8%), and other major students (proportions ranged from 89.1% to 92.0%), respectively (*p* < 0.001) (Table 2).

The negative impacts of negative learning events on girls (mean = 8.66), undergraduate students (mean = 8.25), medical science students (mean = 9.22), and vocational university students (mean = 9.11) were significantly (*p* < 0.001) higher than that of males (mean = 8.27), graduate students (mean = 7.05), other major students (mean score ranged from 8.10 to 8.55), and students of other college types (mean score ranged from 7.93 to 8.48), respectively (Table 2).

### 3.3. Associations between Perceived Academic Stress and Overweight and Obesity among College Students, Stratified by Gender, Grade and College Types

As shown in Table 3, perceived academic stress was significantly associated with increased risk for overweight or obesity for all students (OR = 1.05, 95%CI: 1.00, 1.39), males (OR = 1.09, 95%CI: 1.03, 1.15), undergraduate students (OR = 1.06, 95%CI: 1.00, 1.11), and students from Subordinate universities (OR = 1.13, 95%CI: 1.01, 1.26) after adjusting for covariates. However, these associations were not significant for females, graduates, and students from other types of college.

### 3.4. Associations between Negative Learning Events and Overweight and Obesity among College Students, Stratified by Gender, Grade, and College Type

As shown in Table 4, all students who reported a higher negative learning events score were more likely to be overweight or obese (OR = 1.05, 95% CI: 1.01, 1.09). No significant gender difference was found. The undergraduate students but not graduate students who reported higher negative learning events score were more likely to be overweight or obese (OR = 1.05, 95% CI: 1.01, 1.09). Moreover, the association between negative learning events and overweight or obesity was also significant among students in local universities (OR = 1.07, 95% CI: 1.00, 1.14).

## 4. Discussion

This large-scale study found that both academic stress and negative learning events were positively associated with overweight and obesity among Chinese college students, although the association was modest. Additionally, the study indicated that the association is, to some extent, influenced by some demographic factors: only male students, undergraduates, and students in subordinate universities with higher perceived life stress were more likely to be overweight or obese. The effects of negative learning events on overweight and obesity were only found in undergraduates and students in local universities.

As expected, most of the Chinese college students perceived academic stress, experienced negative learning events, and were impacted by them, which may lead to a series of negative consequences [21]. Consistent with the findings of previous studies [22], this study indicated that female students reported higher levels of academic stress compared to male students. Females may be more sensitive to the stressors; when being in a stressful situation or suffering negative events they may be more easily affected by the problems and more willing to report their emotions [23]. Results of the major differences replicated the previous study which showed that students of medical majors were more academically stressed than students from other majors. Medical students are under much more pressure due to extensive curricula, frequent exams, and strain from working with patients [24]. The study also indicates that undergraduates were more easily likely to encounter negative learning events than graduate students. Graduates may master more professional and time management skills and social support than undergraduates which may help them deal with the negative events and be influenced less.

Psychological stress has been found to be one of the determinants of weight gain through multiple pathways [25]. Biological responses to stress include the impairment of gastrointestinal function that slow gastric emptying and promote energy storage, HPA axis dysregulation, and secretion changes of many biochemical substances that are relevant to weight (e.g., cortisol, leptin, ghrelin and neuropeptide Y) [7]. Moreover, stress can affect behavior such as an increased food assumption and sedentary lifestyle. A study conducted in seven cities of China reported that college students prefer to choose ready-to-eat food and energy-dense foods and increase their food consumption when perceived as having high stress which in turn promotes positive-energy imbalance and increases the risk of obesity [21].

Epidemiological evidence that linked stress as one of the determinants to the development and maintenance of overweight and obesity was accumulating. A cohort study of Hispanic-Latino adults showed that chronic stress is positively related to obesity, with the obese individuals reporting over three chronic stressors [26]. Using data from the Jackson Heart Study, Samson Y et al. showed that African-American adults who perceived higher stress levels were more likely to be obese [27]. The epidemiological data of children and adolescents has also showed an association of higher psychological stress with higher odds of obesity [28]. This study differs in that we focus on the specific academic stressor conducted in Chinese college students, and across these young adults, perceived stress may result in diverse changes in both physiology and behavior compared to children or older adults. In previous studies, multiple stressors often coexist and have greater influence on individuals than one stressor which may explain why the effect of the academic stress on obesity is smaller than prior similar studies [8]. Different measurements of stress and adiposity across the studies may also contribute to the discrepancy. The current study also suggested that a higher incidence and impact of negative learning events may contribute to the likelihood of being overweight and obese. It is possible that people may perceive higher stress after negative events and trigger a stronger stress response which may explain the association [29]. Indeed, negative learning events such as exam failure and financial hardship were often dealt with by avoidance through unhealthy behaviors including overeating, drinking, and smoking more [30]. Although no known researchers have specifically paid attention to negative learning events, the finding concurred with the Dutch study that showed recent stressful life events were positively related to BMI among young adults [31].

Gender-stratified analyses showed that the negative impacts of academic stress on overweight and obesity were more remarkable in male students. The finding is partly consistent with the study in 50 Chinese universities which showed a positive association between perceived life stress and overweight and obesity [9]. Males were found to be more reactive to the stress exposure such as a greater neuroendocrine activation and a high release of cortisol [32]. Moreover, male students are more likely to lose behavior control, eating more and exercising less, while female students seem better able to maintain a healthy lifestyle and a healthy weight when in a stressful situation [33].

Perceived academic stress and negative learning events only affected the risk of obesity for undergraduates, not graduates. The different coping strategies response to stressful events of graduates and undergraduates may explain the difference. It was indicated that undergraduates chose to escape from the plight by internet addiction and overeating, which may lead to overweight or obesity, while graduates master more living skills and prefer to solve the problem in a planned way when being in stressful academic situations [34].

Academic life may account for a significant part of college life for students from subordinate universities and local universities. They may have more excellent academic performance which, in turn, leads to higher expectations, and the failure of exams or worry about further study may have greater influence on them compared to students from other colleges. Hence, the association may be more remarkable than others.

This study has several important strengths. First, the large sample size, about 30,000 college students from different majors, grades, and types of college were recruited in western China which enable us to comprehensively explore the characteristics of academic stress and negative learning events in college students. Another strength was that we analyzed the effects in detail of academic stress and negative learning events on overweight and obesity among college students in different genders, grades, and college types.

The study also has some limitations. First, the cross-sectional design could not address the causal effects of academic stress variables on overweight and obesity among college students. Other factors associated with academic stress (e.g., the intellectual ability and head circumference) [35] and the nutritional status (e.g., dietary intake, physical activity) [36] of college students need to be investigated and considered in future work. It is important to continually monitor changes in academic stress over time alongside changes in nutritional status to observe possible effects of each other in cross-lagged analyses. Second, body weight and height were self-reported, which might have weakened the observed associations. Self-reported height and weight are widely used in population-based studies and are closely correlated to those measured in most cases [37]. This is especially thought to be the case among college students, who pay much attention to their weight status and may measure their weight frequently, and thus may be more likely to report accurately their weight and height. Third, while we observed the association between academic stress and obesity, we did not further explore the mediators of such associations (e.g., food choice, food intake, and physical activities). Future studies may explore the mediators on the associations between academic stress and obesity. Studies that include participants with different educational stages (e.g., primary and secondary school students) are needed to obtain a better understanding of the associations between academic stress and obesity. Longitudinal studies are also warranted to examine the causal relationship of academic stress and negative learning events with overweight and obesity in college students.

## 5. Conclusions

In conclusion, we found that a majority of college students were exposed to high academic stress and negative learning events. Perceived academic stress and negative learning events could increase the risk of being overweight and obese. Interventions are needed to reduce academic stress among college students and may also reduce the prevalence of overweight and obesity.

## Figures and Tables

**Table 1 ijerph-17-05559-t001:** Demographic characteristics of the Chinese college students (Mean (SD)/%) in 2018.

Characteristics	Overall (*n* = 27,343)	Boys (*n* = 11,888)	Girls (*n* = 15,295)	*p*-Value across Gender ^a^
Grade				0.693
Undergraduate	93.4	93.5	93.4	
Graduate	6.6	6.5	6.6	
Major				<0.001
Science and technology	37.1	57.3	21.8	
Medical science	13.8	8.0	18.1	
Agronomy	3.2	3.7	2.8	
Humanities and social science	19.8	10.7	28.4	
Economics and management	18.4	14.8	22.6	
Sports and art	5.7	5.5	6.3	
BMI (kg/m^2^)	20.9 ± 3.9	21.8 ± 4.3	20.1 ± 3.4	<0.001
Overweight and obesity ^c^	13.2	20.8	7.3	<0.001
Academic attainment				<0.001
Below average	28.0	3180 (27.0%)	4348 (28.7%)	
Average	58.9	6529 (55.4%)	9318 (61.6%)	
Above average	13.1	2081 (17.7%)	1464 (9.7%)	
Paternal education level				<0.001
≤Junior middle school	83.6	82.0	84.9	
Senior middle school/vocational schools	14.6	15.9	13.6	
≥College	1.7	2.0	1.5	
Maternal education level				<0.001
≤Junior middle school	86.5	85.3	87.5	
Senior middle school/vocational schools	12.1	13.0	11.4	
≥College	1.4	1.7	1.1	
Household income				
<40,000/year	80.8	78.7	82.5	<0.001
50,000–100,000/year	13.9	14.7	13.3	
>100,000/year	5.3	6.6	4.3	

Abbreviation: BMI: body mass index; Undergraduate include undergraduate students in Grade 1, 2, 3, 4, and 5; Graduate include graduate students in Grade 1, 2, and 3. ^a^: *p*-value was based on Chi-square test for categorical variables and *t*-tests for continuous variables across genders; ^c^: Overweight and obesity were defined by Working Group for Obesity in China for Chinese adults (overweight: Body mass index: 24.0–27.9 kg/m^2^; obesity: Body mass index: ≥28 kg/m^2^).

**Table 2 ijerph-17-05559-t002:** Characteristics of academic stress and negative learning events among college students in China stratified by gender, grade, major, and college type.

All and Subgroups	Perceived Academic Stress			*p* ^a^	Negative Learning Events (Yes/%)	*p* ^a^	Impact of Negative Learning Events ^c^	*p* ^a^
	Low	Medium	High					
All	18.9%	58.2%	22.9%		90.7		8.48 ± 4.39	
Gender				<0.001				<0.001
Male	21.5%	56.1%	22.5%		89.4	<0.001	8.27 ± 4.55	
Female	16.9%	59.8%	23.3%		91.9		8.66 ± 4.24	
Grade ^b^				0.074		<0.001		<0.001
Undergraduate	20.3%	55.9%	23.7%		91.1		8.25 ± 4.28	
Graduate	18.5%	56.3%	25.0%		87.8		7.05 ± 4.36	
Major				<0.001		<0.001		<0.001
Science and technology	19.5%	57.5%	23.0%		90.7		8.35 ± 4.40	
Medical science	14.5%	61.9%	23.6%		92.0		9.22 ± 4.27	
Agronomy	21.6%	56.4%	22.0%		89.8		8.10 ± 4.32	
Humanities and social science	16.4%	57.0%	26.5%		92.1		8.51 ± 4.30	
Economics and management	21.2%	58.5%	20.3%		90.1		8.55 ± 4.40	
Sports and art	22.0%	57.9%	20.2%		89.1		7.88 ± 4.37	
College type				0.308		0.199		<0.001
Subordinates universities	22.4%	55.9%	24.7%		91.5		7.93 ± 4.15	
Local universities	19.5%	57.0%	23.4%		91.0		8.36 ± 4.31	
Private universities	18,4%	60.3%	21.3%		90.3		8.48 ± 4.39	
Vocational universities	17.1%	61.1%	21.8%		91.4		9.11 ± 4.39	

Variable definition: Score of “impact of negative learning events” was calculated by summing the scores of the four items in the scale, and higher score more negative impact of negative learning events; ^a^: *p*-value was based on Chi-square test for perceived academic stress and negative learning events across genders, grade, major, and college type. ^b^: The difference of academic stress between graduate and undergraduate was only compared among students from Subordinate and local universities (sample size: 14,494), as only these universities had graduate students. ^c^: The participants were asked to rate the degree to which the four negative learning events affected them using a 5-point Likert scale where “never = 1” to “extremely heavy = 5”. The average scores of four events were calculated to indicate the negative impacts of the negative learning events, with higher score indicating more negative events.

**Table 3 ijerph-17-05559-t003:** Mixed-effects model for the associations between perceived academic stress and overweight and obesity among college students in China, stratified by gender, grade, and college type (*n* = 27,343).

All and Subgroups	Overweight and Obesity		
	OR	95%CI	*p*
All ^a^	**1.05**	**(1.00, 1.10)**	**0.039**
Gender ^b^			
Male	**1.09**	**(1.03, 1.15)**	**0.002**
Female	0.949	(0.87, 1.03)	0.235
Grade ^c^			
Undergraduate	**1.06**	**(1.00, 1.11)**	**0.032**
Graduate	0.98	(0.84, 1.14)	0.753
College type ^d^			
Subordinates universities	**1.13**	**(1.01, 1.26)**	**0.038**
Local universities	1.02	(0.94, 1.10)	0.659
Private universities	1.07	(0.95, 1.22)	0.266
Vocational universities	1.04	(0.96, 1.14)	0.331

Overweight and obesity were defined by Working Group for Obesity in China for Chinese adults (overweight: Body mass index: 24.0–27.9 kg/m^2^; obesity: Body mass index: ≥28 kg/m^2^); Non-overweight = 0, overweight or obesity = 1. Linear/logistic mixed-effects model was used to analyze the associations between perceived academic stress and weight status. ^a^: The mixed-effects model adjusted gender, grade, major, college type, father and mother education level, household income, and academic attainment. ^b^: The mixed-effects model adjusted grade, major, college type, father and mother education level, household income, and academic attainment. ^c^: The mixed-effects model adjusted gender, major, college type, father and mother education level, household income, and academic attainment. ^d^: The mixed-effects model adjusted gender, major, grade, father and mother education level, household income, and academic attainment. Numbers in bold indicate statistical significance.

**Table 4 ijerph-17-05559-t004:** Mixed-effects model for the associations between negative learning events and overweight and obesity among college students in China, stratified by gender, grade, and college type (*n* = 27,343).

All and Subgroups	Overweight and Obesity		
	OR	95% CI	*p*
All ^a^	**1.05**	**(1.01, 1.09)**	**0.009**
Gender ^b^			
Male	1	-	-
Female	1.02	(0.95, 1.09)	0.608
Grade ^c^			
Undergraduate	**1.05**	**(1.01, 1.09)**	**0.011**
Graduate	1.05	(0.94, 1.19)	0.352
College type ^d^			
Subordinates universities	1.05	(0.96, 1.16)	0.228
Local universities	**1.07**	**(1.00, 1.14)**	**0.022**
Private universities	1.04	(0.95, 1.14)	0.411
Vocational universities	1.05	(0.98, 1.11)	0.155

Overweight and obesity were defined by Working Group for Obesity in China for Chinese adults (overweight: Body mass index: 24.0–27.9 kg/m^2^; obesity: Body mass index: 28 kg/m^2^); Non-overweight = 0, overweight or obesity = 1. Linear/logistic mixed-effects model was used to analyze the associations between negative learning events and weight status. ^a^: The mixed-effects model adjusted gender, grade, major, college type, father and mother education level, household income, and academic attainment. ^b^: The mixed-effects model adjusted grade, major, college type, father and mother education level, household income, and academic attainment. ^c^: The mixed-effects model adjusted gender, major, college type, father and mother education level, household income, and academic attainment. ^d^: The mixed-effects model adjusted gender, major, grade, father and mother education level, household income, and academic attainment. Numbers in bold indicate statistical significance.

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
