# Peer review of "Higher Academic Stress Was Associated with Increased Risk of Overweight and Obesity among College Students in China"

_ijerph, 2020, doi:10.3390/ijerph17155559_

Round 1

Reviewer 1 Report

The study is of interest to the research field. The authors have carried out an adequate and in-depth study, with results that seem significant and by the object of the study.

However, the following comments are provided to improve the quality of the document:The Introduction lacks some more information on the effects of high levels of stress on university students. How stress impacts academic performance and previous studies that verify these events.
Similarly, it seems necessary and appropriate to include authors who have studied academic stress, the most neural part, and its effects during the teaching and learning process.

It would be interesting if the authors included a section on "Future lines of research", to consider the scope of possible extensions to other educational stages or including other study variables.

Author Response

  1. The Introduction lacks some more information on the effects of high levels of stress on university students.

Response: We have added related information on the effects of high levels of stress on university students in the second paragraph in Introduction. (L45-L46)

  1. How stress impacts academic performance and previous studies that verify these events.
    Similarly, it seems necessary and appropriate to include authors who have studied academic stress, the most neural part, and its effects during the teaching and learning process.

Response: Thanks for the comments. However, in this study, we focused on the academic stress and its associations with overweight and obesity. We have rationalized the hypothesized associations between academic stress and overweight and obesity among college students, and why we focused on the academic stress of college students in the context of China. Therefore, we did not report the impacts of stress on academic performance.  

  1. It would be interesting if the authors included a section on "Future lines of research", to consider the scope of possible extensions to other educational stages or including other study variables.

Response: We have added “Future lines of research” in the last paragraph in the Discussion. (L336- L344)

Please see the revised manuscript in the  attachment.

Reviewer 2 Report

  1. I do not understand at the end of all the students were university or were boys or girls.
  2. I think that it is not valid to draw conclusions about weight or obesity. Using the self- reported BMI body mass index. Why didn't they consider other additional measurements like waist, hip and neck circumference?.
  3. The physical measurements they made in this work are often reported for convenience, and particularly because the study is on large samples. Surely as researchers they did not visit the participants for the measurements.
  4.  I think that both men and women can underestimate their height and especially their weight.

Author Response

  1. I do not understand at the end of all the students were university or were boys or girls.

Response: Thanks. A total of 27, 343 college students participated in this cross-sectional study in 2018, of which 11,888 were boys and 15,295 were girls. In Table 2, Table 3, and Table 4, for the analyses in all students, the sample included both boys and girls. For gender stratified analyses, the sample included boys or girls. For grade- and major- stratified analyses, the sample included both boys and girls.

  1. I think that it is not valid to draw conclusions about weight or obesity. Using the self- reported BMI body mass index. Why didn't they consider other additional measurements like waist, hip and neck circumference?

Response: Thanks. The measurements such as waist, hip and neck circumference were not measured in this study. BMI is a universally used measurement to define overweight and obesity in children and adults. Due to the large sample size in this epidemiology study, the body weight and height were self-reported. We have indicated this as a limitation in the study. However, the participants were college students, they may pay much attention to their weight status and measure their weight frequently, and thus may be more likely to report accurately their weight and height (L331-L336). 

  1. The physical measurements they made in this work are often reported for convenience, and particularly because the study is on large samples. Surely as researchers they did not visit the participants for the measurements.

Response: Thanks. We did not measure body weight and height objectively, because the study is on large samples. All the students completed the self-administered questionnaire in classroom settings.  

  1. I think that both men and women can underestimate their height and especially their weight.

Response: Thanks. We have indicated this a limitation in the study. (L331-L336) 

Please see the revised manuscript in the attachment.

Reviewer 3 Report

This study examined associations between academic stress and weight status among college students what I found very interesting topic, especially in cultural context of Chine where students perceive high academic pressure due to their future success in labour market. Results of this article might provide important information not only for educational system, as overweight and obesity is associated with many negative health consequences in adulthood. Moreover, I highly appreciate a large representative sample of students. However, some issues need to be solved before publication.

Title – sample size in title is redundant and I would recommend revise a title to make it more attractive for readers. “Associations between academic stress and weight” is vague and it could be revised following main results of the study.

Introduction -  I wonder if there are some more information about attitudes towards school, education or support from teacher/schoolmate among students in China as it might be significant contributors of perceived stress in schools. If yes, description of prevalence or associations would be beneficial for supporting aim of study and why is important to focus on this topic.

P2L60 –Investigation characteristics of academic stress is redundant. I would skip this sentence and clearly state moderation effect of demographic factors. Please, replace term “determine” by “examine”.

Moreover, when authors would like to examine moderation effect of demographic factors on association between academic stress and overweight, one paragraph in introduction should be focused on recent evidence about this topic.

Materials and Methods -   study design and participants – please, provide information about administration of questionnaire (paper questionnaire or online data collection, who administrated questionnaire) and information about sample (min-max age, mean age, % boys)

 Variables and measurements – as outcomes were chosen two variables: BMI and overweight/obesity. Here I recommend to authors to choose one of them based on aim and title of article, as using both of them is redundant. Moreover, as in whole article authors work with overweight and obesity, I would incline to use this variable in statistical analysis.

Exposure variables – perceived academic stress – description of this item should be revised a little bit. Specifically, L96-98 should indicate that response categories were categorised into three categories: low stress, medium stress, high stress. Moreover, prevalence of this variable is not included in description table 1. Similarly, in description negative learning events should be mentioned that students who answered that these events happened in the past 6 months were asked to rate degree to which these events affected them. As I understand it correctly, authors calculated sum score, I would recommend use this term in spite of average scores.

Statistical analysis and results – here I have several comments to make this part clearer and more readable. First of all, as I mentioned earlier, authors should choose one of outcomes, and based on aim of this study I incline to dichotomized variable overweight/obesity, as using both BMI and overweight/obesity is redundant. Furthermore, set of chi square and t-test analysis are redundant as the same association are assess in logistic regression model. Similarly, gender, grade and college type stratified analyses are redundant as the very same could and should be done assessing interaction of explored variables with gender, grade and college type on their effect on outcome measures. And I would choose only one moderator based on literature and authors expectations. In abstract is mentioned that authors examined moderating effect of demographic factors but nowhere further in text is not mentioned that authors do that, neither results connected to moderation effect.  

Then statistical analysis should be described in more details, e.g. analysing crude effect or crude effect adjusted on covariates, using enter or stepwise procedure, interaction of exposure variables with covariates, e.g. gender on outcome measure.

In Results and Table 1 part is mentioned academic achievement/attainment, but it is not described in methods part. Please, check if it is the same and it is mistake in English as well as if it should be added into methods part.

After implementing comments connected to statistical analysis, results part and discussion should be revised.

Tables – After implemented comments tables have to be revised. I have small technical comments to lay out of tables. To make them more readable, I recommend to align “characteristics” column to left side and add space after every variable. In footnote should be explained only abbreviation used in the table. Every comment connected to description of analysis should be mentioned in statistical analysis part, e.g. P4L141-143.      

Author Response

  1. Title– sample size in title is redundant and I would recommend revise a title to make it more attractive for readers. “Associations between academic stress and weight” is vague and it could be revised following main results of the study.

Response: The title was changed to “Higher academic stress was associated with increased risk of overweight and obesity among college students in China”. (L2-L4)

  1. Introduction- I wonder if there are some more information about attitudes towards school, education or support from teacher/schoolmate among students in China as it might be significant contributors of perceived stress in schools. If yes, description of prevalence or associations would be beneficial for supporting aim of study and why is important to focus on this topic. 

Response: Thanks. We have mentioned “In China, students’ academic excellence has been a social criterion and even become the only standard for Chinese parents to judge their children. Rapid social changes and implementation of the one-child policy have contributed to elevate academic and job competition and directly increase college student’s academic pressure” in the Introduction. (L67-L73)

  1. P2L60 –Investigation characteristics of academic stress is redundant. I would skip this sentence and clearly state moderation effect of demographic factors. Please, replace term “determine” by “examine”.

Response: We have revised the research aims according to the comments (L75-L85).

  1. Moreover, when authors would like to examine moderation effect of demographic factors on association between academic stress and overweight, one paragraph in introduction should be focused on recent evidence about this topic.

Response: We have added the rational of the moderation effects of demographic factors (gender, grade, and major) in Introduction. (L51-57)

  1. Materials and Methods-   study design and participants – please, provide information about administration of questionnaire (paper questionnaire or online data collection, who administrated questionnaire) and information about sample (min-max age, mean age, % boys)

Response: We have added the detail of administration of questionnaire and % boys in the Materials and Method part. However, participants’ age was not measured in this study, so we could not add, we use grade to indicate their age. (L96-L104)

  1. Variables and measurements– as outcomes were chosen two variables: BMI and overweight/obesity. Here I recommend to authors to choose one of them based on aim and title of article, as using both of them is redundant. Moreover, as in whole article authors work with overweight and obesity, I would incline to use this variable in statistical analysis.

Response: We have deleted the analyses on BMI, and only focused on overweight and obesity in the manuscript.     

  1. Exposure variables – perceived academic stress – description of this item should be revised a little bit. Specifically, L96-98 should indicate that response categories were categorised into three categories: low stress, medium stress, high stress. Moreover, prevalence of this variable is not included in description table 1. Similarly, in description negative learning events should be mentioned that students who answered that these events happened in the past 6 months were asked to rate degree to which these events affected them. As I understand it correctly, authors calculated sum score, I would recommend use this term in spite of average scores.

Response: The descriptive of the three categories of perceived academic stress have been revised accordingly (L124-L126). The prevalence of this variable is included in the Table 2. The description of negative learning events has been revised according (L133-L134). The average score of the four items were calculated and used in this study.

  1. Statistical analysis and results– here I have several comments to make this part clearer and more readable. First of all, as I mentioned earlier, authors should choose one of outcomes, and based on aim of this study I incline to dichotomized variable overweight/obesity, as using both BMI and overweight/obesity is redundant.

Response: The reply was the same as above.

  1. Furthermore, set of chi square and t-test analysis are redundant as the same association are assess in logistic regression model.

Response: The results of chi square and t-test analysis in Table 2 did not adjust for covariates. However, the associations presented in Table 3 adjusted for covariates. Therefore, they are not the same.

  1. Similarly, gender, grade and college type stratified analyses are redundant as the very same could and should be done assessing interaction of explored variables with gender, grade and college type on their effect on outcome measures. And I would choose only one moderator based on literature and authors expectations. In abstract is mentioned that authors examined moderating effect of demographic factors but nowhere further in text is not mentioned that authors do that, neither results connected to moderation effect.  

Response: Thanks. As these potential stratifying variables (i.e., gender, grade, and college type) were categorical variables, we prefer to use stratified analyses to test the potential modifying effects. Based on findings of previous literature, we thought gender, grade, and college type were potential modifier of such associations. In the abstract and whole manuscript, we clarified to test the modifying effects of gender, grade, and college type.        

  1. Then statistical analysis should be described in more details, e.g. analysing crude effect or crude effect adjusted on covariates, using enter or stepwise procedure, interaction of exposure variables with covariates, e.g. gender on outcome measure.

Response: We have revised the statistical analysis accordingly. (L149-L161)

  1. In Results and Table 1 part is mentioned academic achievement/attainment, but it is not described in methods part. Please, check if it is the same and it is mistake in English as well as if it should be added into methods part.

Response: We have added the academic attainment in the covariates in Methods part.  (L143-L144) 

  1. After implementing comments connected to statistical analysis, results part and discussion should be revised.
    Response: We have revised.
  2. Tables– After implemented comments tables have to be revised. I have small technical comments to lay out of tables. To make them more readable, I recommend to align “characteristics” column to left side and add space after every variable. In footnote should be explained only abbreviation used in the table. Every comment connected to description of analysis should be mentioned in statistical analysis part, e.g. P4L141-143. 

Response: We have revised all the Tables.

Please see the revised manuscript in the attachment.

Reviewer 4 Report

The manuscript is interesting and novel. The structure and methodology used are adequate. The introduction is adequate, the results are interesting and support the discussion. This type of study considers educational and health aspects, which is of interest to researchers. However I have the following comments.

Major Comments:
1. In the introduction, I suggest including dietary aspects related to body weight gain and obesity, especially high energy intake, refined carbohydrates, total and saturated fat.
Suggested reference:
Relevant Aspects of Nutritional and Dietary Interventions in Non-Alcoholic Fatty Liver Disease. Int J Mol Sci. 2015; 16: 25168-98.

2. Academic stress, school performance as well as nutritional status depends on multiple factors. In this regard, in the discussion the authors must be very specific in this regard. Therefore I suggest writing a short paragraph about it.
Suggested reference:
A multifactorial approach of nutritional, intellectual, brain development, cardiovascular risk, socio-economic, demographic and educational variables affecting the scholastic achievement in Chilean students: An eight-year follow-up study. PLoS One. 2019; 14: e0212279.

3. To have a better understanding of the effect it would be very good to include a multiple regression analysis.

Minor Comments:
1. In the title it is not necessary to include the number of subjects evaluated
2. In keywords it is not necessary to use the word China
3. Improve the writing of the study objective

Author Response

Major Comments:
1. In the introduction, I suggest including dietary aspects related to body weight gain and obesity, especially high energy intake, refined carbohydrates, total and saturated fat.
Suggested reference:
Relevant Aspects of Nutritional and Dietary Interventions in Non-Alcoholic Fatty Liver Disease. Int J Mol Sci. 2015; 16: 25168-98.

Response: In the part of introduction, we had listed such evidence that a laboratory study reported that college students took much more calories, carbohydrates and sugars when being academically stressed, which may increase the risk of overweight and obesity. (L60-L62)

  1. Academic stress, school performance as well as nutritional status depends on multiple factors. In this regard, in the discussion the authors must be very specific in this regard. Therefore I suggest writing a short paragraph about it.
    Suggested reference:
    A multifactorial approach of nutritional, intellectual, brain development, cardiovascular risk, socio-economic, demographic and educational variables affecting the scholastic achievement in Chilean students: An eight-year follow-up study. PLoS One. 2019; 14: e0212279.

Response: Thanks for the insightful comments. There are multiple factors associated with college students’ academic stress and nutritional status, however, we did not measure these variables in this study. Therefore, we indicated these as limitations in the Discussion and cited the recommended references (L329-L335). (Reference 35 and Reference 36)

  1. To have a better understanding of the effect it would be very good to include a multiple regression analysis.

Response: Considering the cluster effects, we preferred to use linear/logistic mixed-effects model to than multiple regression analysis to analyze the associations of academic stress and negative learning events with overweight and obesity.  

Minor Comments:
1. In the title it is not necessary to include the number of subjects evaluated

Response: We have deleted the number of subjects in the title and revised the title. (L2-L4)

  1. In keywords it is not necessary to use the word China

Response: We have deleted the key word “China”.

  1. Improve the writing of the study objective

Response: We have revised the study objective (L75-L85).

Please see the revised manuscript in the attachment.

Reviewer 5 Report

Dear authors,

Their work is well structured for which I congratulate them. However, I made some comments so that you can assess them:

  • Literature: I think it is necessary to incorporate reference between 2019 and 2020. 
  • Conclusions: Can you develop more?

Best regards. 

Author Response

Conclusions: Can you develop more?

Response: We have updated the references, and have added references between 2019 and 2020 (Reference 7,Reference 12, Reference 29 and Reference 35).

Please see the revised manuscript in the attachment.

Round 2

Reviewer 2 Report

Accept in present form

Reviewer 3 Report

Authors made a big progress with this paper and I recommend it for publication.